# Structural Optimization of Vertically-Stacked White LEDs with a Yellow Phosphor Plate and a Red Quantum-Dot Film

**DOI:** 10.3390/nano12162846

**Published:** 2022-08-18

**Authors:** Seung Chan Hong, Jae-Hyeon Ko

**Affiliations:** School of Nano Convergence Technology, Nano Convergence Technology Center, Hallym University, Chuncheon 24252, Korea

**Keywords:** quantum dot, LED, color rendering index, remote-type, phosphor

## Abstract

A remote-type white light-emitting diode (LED) consisting of a red quantum-dot (QD) film and a yellow phosphor plate was studied by both experiment and optical simulation. The sequence of the two color-conversion films had a substantial effect on the color-rendering properties of the vertically-stacked white LED, and the optimized configuration exhibited a high color rendering index of more than 90 thanks to the enhanced red component via the QD film. For the design of high-power white LED devices of a remote type, it was necessary to locate the color-conversion films below the diffuser plate to remove the substantial color dispersion depending on the viewing angle. The present study shows that high power and high color-rendering white LED devices can be realized in terms of two vertically-stacked color-conversion materials, which would provide long-term stability due to the remote design.

## 1. Introduction

White light-emitting diodes (W-LEDs) have been spreading in both display technologies and general lighting applications. W-LEDs usually consist of GaN-based blue LED chips and color-conversion materials, such as phosphors and quantum dots (QDs) [1]. Because of the cost issue, conventional W-LEDs have been adopting yellow phosphors as the sole color-conversion material, the Ce-doped yttrium aluminum garnet (YAG, Y_3_Al_5_O_12_) being the most representative one [2]. In this case, the emitting spectrum consists of a sharp blue peak from the blue LED chip and a broad yellow peak emitted from the YAG phosphor. This approach has several advantages such as high efficiency, low cost, simple fabrication process, etc. However, the emitting spectrum lacks deep red spectral components which makes the W-LEDs have a low color rendering index (CRI). This issue has become more important these days because people tend to spend more time in their houses due to COVID-19, and, thus, color-rendering characteristics of lighting devices have become more important for indoor life. 

There have been several ways to improve the color-rendering properties of W-LEDs, such as including both green and red phosphors (or yellow and red phosphors) to enhance the deep red components in the emitting spectrum and, thus, to improve the CRI [3,4]. QDs, which are nanometer-sized semiconductors, may be an alternative to phosphor materials as a color conversion material [5,6]. The Quantum confinement effect of QDs plays a critical role in their emitting properties because the main emitting wavelength of QD is determined by its size [7,8,9]. Therefore, the emitting color from QDs can be easily controlled in terms of their sizes without changing the chemical compositions. Because of the high color purity of QDs, they have first been adopted in display technologies, such as LCD (Liquid Crystal Display) backlights and color-conversion materials in RGB subpixel structures [10,11,12,13,14,15,16]. 

Another issue of W-LEDs is the long-term stability of color-conversion materials. which are affected by the high thermal load caused by hot LED chips. High temperature increases the probability of nonradiative transitions in phosphors and QDs, resulting in lower luminous efficiency and a shorter lifetime of W-LEDs [17,18,19]. In addition, the transparent epoxy resin, in which color-conversion materials are dispersed homogeneously, may degrade when it is exposed to high temperatures for a long time. Several remote designs of phosphors have been adopted to solve this problem, where the remote phosphor is placed over the blue LED chips with a certain distance between them [20,21,22,23,24,25]. QD can also be designed as a remote component to avoid degradation [26,27,28,29,30,31,32,33]. For example, QD particles were incorporated in polymers or glasses as composites in various forms [34,35,36,37,38,39,40,41,42,43,44,45,46,47]. In addition, large-size QD films or QD caps could be developed for practical applications in displays and illumination [48,49,50]. These designs could improve the efficiency, color-rendering characteristics, and long-term stability of W-LEDs successfully. However, a more detailed investigation is necessary for the optimization of W-LEDs consisting of at least two color-conversion materials adopted in a remote design. Especially, the order of the color-conversion materials and their positions in W-LEDs need to be investigated in terms of experiment and simulation from the viewpoint of key lighting performances. The present study is aimed at optimizing vertically-layered W-LEDs where two remote components, a yellow phosphor plate and a red QD film, were placed over blue LED chips with or without transparent epoxy resin. Especially, the effect of the remote configuration on the color-rendering characteristic was studied in detail. In addition, a high-power downlighting W-LED luminaire in a remote design was also studied by ray-tracing simulation based on the above optimization. 

## 2. Materials and Methods

A typical blue LED package (IWS-L5056-UB-K3, Itswell Co., Incheon, Korea) with a dimension of 5.4 × 5.0 × 1.6 mm^3^ was used as an excitation source for phosphors and QDs. YAG plates with a Ce concentration of 0.5 ± 0.02% and an area of 5.0 × 5.0 mm^2^ were purchased with three thicknesses of 0.1, 0.15, and 0.2 mm (Baikowski Japan, Narashino, Japan). Appendix A includes the photo of the used YAG phosphor plates. The PL (photoluminescence), absorption, and transmission spectra were measured by using a fluorescence and absorbance spectrometer (Duetta, Horiba Co., Kyoto, Japan), and the results are also shown in Appendix A. It shows that the absorption band is maximum at the emitting wavelength of the blue LED chip (~460 nm). Both surfaces were polished to optical quality. 

CdSe/ZnS core-shell QDs were fabricated by using the conventional hot injection method. The average diameter of the grown QDs was approximately 6 nm. For homogeneous dispersion, QD particles were mixed with irregular hollow silica (SG-HS40, Sukgyung At Co., Ansan, Korea), the approximate size of which was ~40 nm. These particles were mixed with the triazine epoxy resin, which was coated on a PET (polyethylene terephthalate) substrate in terms of the roll-to-roll slot die method. The details of the fabrication of the QD films are included in Ref. [48]. Appendix A includes the photo of the fabricated red QD film, PL (photoluminescence), absorption, and transmission spectra. The PL spectrum shows a peak at the wavelength of ~630 nm. 

Four designs of the vertically-stacked structure were studied experimentally as shown in Figure 1. The “YR” structure indicates that the yellow phosphor is placed below the red QD film, whereas the “RY” structure shows the reversed case, i.e., the red QD film is put below the yellow phosphor plate. In these cases, no epoxy resin fills the space between the blue LED chips and the two remote color-conversion components. If this space is filled with transparent photopolymer, the two structures will be denoted as “YRe” and “RYe”, where “e” denotes epoxy. Transparent epoxy is usually formed over LED chips to protect them for long-term operation. A typical photopolymer (NOA63, Norland Co., Jamesburg, NJ, USA) was used as an epoxy, which was hardened by using the ultraviolet lamp (S30365FL, Skycares Co., Gimpo, Korea) for 20 min. Photometric quantities (luminous efficiency, intensity, luminance, etc.), the color coordinates, the emitting spectrum, and the CRI were measured by using a goniophotometer (LED626, Everfine Co., Hangzhou, China), a spectroradiometer (PR-670, Photo Research Co., Chatsworth, CA, USA), and an illuminance meter (SPIC-200A, Everfine Co., Hangzhou, China). 

The same structures were modeled in the ray-tracing simulation. Commercially-available software (LightTools, ver.9.1.0, Synopsys Co., Mountain View, CA, USA) was used for the simulation. Figure 2 shows the LED package and the vertically-stacked W-LED structures. The LED package had the dimensions of 5 × 5 × 1.6 mm^3^. The diameter of the bottom and the upper opening were 3.5 and 4.55 mm, respectively. A blue LED chip with a refractive index of 2.4 and dimensions of 0.2 × 0.2 × 0.01 mm^3^ was placed at the center on the bottom surface. The emitting spectrum of the blue LED had a peak wavelength of 450 nm with a full width at half maximum(FWHM) of 25 nm, similar to that in the experiment. The emitting distribution was Lambertian. The LED frame was PMMA(Polymethyl methacrylate) with a refractive index of 1.4936. A reflection layer having an absorption of 15% and a reflectance of 85% with a diffuse Gaussian reflection distribution (Gaussian angle of 15°) was formed on the frame. All other dimensions were the same as those in the experiment. The details of the simulation conditions are shown in Table 1, Table 2 and Table 3.

The YAG phosphor plate was modeled as phosphor particles with a diameter of 23 μm and a refractive index of 1.8. These particles were homogeneously dispersed in ceramic MgAl_2_O_4_ with a refractive index of 1.7188 at 16 wt%. The thickness of the phosphor plate was adjusted in a range of 0.12~0.21 mm. The red QD film was modeled as QD particles, which were dispersed in the PET (Polyethylene terephthalate) with a mean free path (MFP) of 0.1 mm. The emitting spectrum of QD had a peak wavelength of 632 nm with a FWHM of 30 nm. The thickness of the QD film was 30 μm. The distance between the YAG phosphor plate and the red QD film was set to be 0.01 mm, mimicking the existence of an air gap between the two components used in the experiment. The PL and absorption spectra of YAG phosphors and QDs obtained from the experiment were used in the simulation, whereas the built-in excitation spectrum of YAG phosphors and the quantum yield of the red CdSe/ZnS QDs in the software were used. 

To investigate the possibility of realizing and optimizing remote-type high-power LED lighting, various configurations were studied, as shown in Figure 3. The square LED lighting fixture consists of 100 blue LEDs with a pitch of 20 mm and an array of 10 × 10, as shown in Figure 3a. All dimensions of the lighting fixture are shown in Figure 3a, where the fixture frame was set to be aluminum which plays the role of a mirror reflector. Figure 3b exhibits eight different configurations, where Y, R, and D denote the yellow phosphor film, the red QD film, and a diffuser plate, respectively. All matrix materials were PET. The YAG particles were included in the yellow phosphor film at 50 wt%. The MFP of QD particles in the film was 0.12 mm. The “mix” indicates the film in which the yellow phosphor and red QD particles were randomly mixed at the same concentrations used in the individual film. The dimensions of the three color-conversion films were 368 × 368 × 0.1 mm^3^. In the diffuser plate, whose dimensions were 368 × 368 × 2 mm^3^, TiO_2_ spherical particles at the center radius of 225 nm with a FWHM of 155 nm were homogeneously dispersed in the host material (polycarbonate) at 0.1 wt%. The PL, absorption, excitation spectra, and quantum yield in addition to the size distributions of phosphors and TiO_2_ particles are included in Appendix A. 

## 3. Results and Discussion

Figure 4 shows the emitting spectra of four W-LEDs shown in Figure 1 at three different thicknesses of the yellow phosphor plate. In the cases of RY and RYe configurations, the peak heights of the red component near 630 nm are substantially high compared to the YR and YRe configurations. In the former, the blue excitation light from LED chips excites the red QD film first, after which a part of it may hit the yellow phosphor plate for further excitation. Since the red light from the QD film is beyond the absorption band of the yellow phosphor plate, the red light from the QD film can easily pass through the yellow phosphor plate and contribute to the emitting spectrum. On the other hand, the YR and YRe configurations display strong and broad yellow peaks whereas the red peak from the QD film is very weak. It indicates that the amount of blue light remaining after passing through the yellow phosphor plate is not enough to excite red QDs in the film. The height of the blue peak decreases substantially as the thickness of the phosphor plate increases reflecting an increasing color-conversion efficiency. One more interesting result shown in Figure 4 is that the red peak becomes stronger as the thickness of the yellow phosphor plate increases. As the phosphor thickness increases, the portion of the light reflected from the phosphor plate toward the red QD film increases, which further excites the red QDs, resulting in a higher red peak. 

Figure 5 shows the dependence of the luminance on the thickness of the phosphor plate for four configurations. Since the power consumption of the blue LED chips was maintained as constant, the luminance value is linearly proportional to the luminous efficiency. The YR structure shows the highest luminance among the four configurations irrespective of the phosphor thickness. The luminance is sensitive to the spectral components of the emitting spectrum and becomes large when the greenish-yellow component is strong where the photopic response becomes maximum. In this context, the YR configuration is more favorable than the RY configuration due to its stronger yellow component in the emitting spectra, which is indeed observed in Figure 4. On the other hand, the YR and RY configurations exhibit higher luminance values than the YRe and the RYe ones do, respectively. When the epoxy resin fills the space in the LED package as shown in Figure 1b, the blue light emitted from the LED chip is partially reflected at the resin-air interface via the total internal reflection (TIR), resulting in a lower excitation probability of the phosphor and QD materials. 

Another characteristic obtained from Figure 5 is that the efficiency becomes the largest at the intermediate thickness of 0.15 mm. When the YAG plate becomes thicker at the initial stage, the color-conversion probability increases contributing to the high luminous flux and, thus, the luminous efficiency increases as well. However, the efficiency decreases at a much thicker phosphor plate due to excessive scattering events in the phosphor plate, which hinders the escape of the light generated in the plate. This leads us to conclude that the appropriate thickness of the YAG phosphor plate at the present Ce concentration is around 0.15 mm. 

Figure 6 shows the change in the color coordinates of the four configurations under three different phosphor thicknesses on the CIE1931 chromaticity diagram. The color coordinates can be grouped into two categories, YR/YRe and RY/RYe. As the red spectral component is stronger in the RY/RYe configurations compared to the YR/YRe configurations, their color coordinates are located closer to the red region on the chromaticity diagram. In addition, the color coordinates of all cases shift to the upper right direction toward the yellow region as the thickness of the phosphor plate increases indicating larger color conversion in the thicker phosphor plate. This trend is clear from the change in the correlated color temperature (CCT). The CCT is in the range of 6400~7600 K with the thinnest phosphor plate (0.1 mm) and decreases with increasing phosphor thickness due to enhanced color conversion in the thicker phosphor plates. The RY/RYe configurations exhibit the lowest CCT values of ~3000 K due to the stronger red components in the emitting spectra. The modification of the emitting spectrum has a pronounced effect on the color-rendering properties. As the phosphor plate becomes thicker in the RY and RYe configurations, both the yellow and the red peaks become stronger, and the CRI values are more than 90 when the thickness is 0.2 mm. For this high color-rendering performance, it is important to complement the red spectral region, which is insufficient in conventional W-LEDs, in terms of red QDs. It is important to increase both green (yellow) and red spectral components evenly to secure high color-rendering properties. 

Optical characteristics of the vertically-stacked W-LED models shown in Figure 2 were investigated by using the ray-tracing technique. Figure 7 shows the emitting spectra of the four LED configurations when the phosphor thickness was 0.14 mm in the simulation model. Similar to the spectra in Figure 4, YR/YRe and RY/RYe configurations display quite different emitting spectral characteristics. RY/RYe configurations show distinct red peaks at ~630 nm due to the strong excitation of the red QD film, whereas YR/YRe configurations exhibit more enhanced yellow peaks from the YAG phosphor peak located near the excitation source (blue LED). It is also noticed that filling the gap between the blue LED and the color-conversion material with a transparent epoxy slightly decreases the spectrum and thus the efficiency. The total internal reflection at the epoxy-air interface returns part of the excitation light toward the blue LED chips, which decreases the color-conversion efficiency compared to the case where no epoxy is used. One remark on the spectrum is that the red peaks in the emitting spectra of RY and RYe configurations are lower than those of the corresponding experimental configurations. The main origin of this difference is directly related to the MFP of red QDs in the film. If the MFP is finely adjusted, the emitting spectrum of the simulation would be more like that of the experiment.

Figure 8 shows the change in the color coordinates of the four configurations as the phosphor thickness increases from 0.12 to 0.21 mm in the simulation model. The overall behaviors are similar to the experimental results shown in Figure 6. The color coordinates can be grouped into either YR/YRe or RY/RYe configurations and shift to the yellow region as the phosphor thickness increases. The order of the two vertically stacked color-conversion materials is important in the color characteristics. 

The efficiency is sensitive to the phosphor thickness as suggested by the experimental result shown in Figure 5. The dependences of the luminous flux, the luminous intensity, and the luminance of the white LEDs on the phosphor thickness are shown in Appendix A for the four configurations. It shows that the efficiency decreases monotonically with increasing phosphor thickness from 0.12 to 0.21 mm. As the phosphor plate becomes thicker, the multiple scattering in the plate becomes more substantial, resulting in higher probability of light trapping and, thus, lower efficiency. It also indicates that the optimized phosphor thickness of the simulation model is located below 0.12 mm where the color-conversion efficiency and the multiple scattering of the generated light may be compromised for the highest efficiency. The difference in the optimal thickness between the experiment and simulation is because the simulation conditions are not the same as with the experimental conditions. The phosphor plate used in the experiment is a ceramic-type where grain-boundary effect is also included, which is difficult to include in the simulation model. In addition, the CCT changes from ~6440 K at the thickness of 0.12 mm to ~4000 K at 0.21 mm. This is due to higher color-conversion probability and, thus, a larger portion of the yellow peak in the whole emitting spectrum at larger phosphor thickness. It suggests that choosing an appropriate thickness of the yellow phosphor is one way to adjust CCT and other color properties of the lamp. 

The simulation results for the vertically-stacked W-LED models shown in Figure 7 and Figure 8 are consistent with those obtained from the experiment, suggesting that the present simulation model is reliable for further studies. In this context, it is desirable to study high-power downlighting structures consisting of two remote-type color-conversion films by optical simulation. It is important to adopt high-power LED lighting in museums, exhibition halls, and industrial fields where high color-rendering performances are required. All configurations shown in Figure 3 were studied by optical simulation. Figure 9 shows the emitting spectra of all configurations. The red peak near 630 nm becomes more pronounced for the configurations where the red QD film is below the yellow phosphor film or the mixed film is adopted on the diffuser plate. On the other hand, the broad yellow peak is more substantial for the configurations where the yellow phosphor film is located below the red QD film. This is very similar to the results obtained from the small-size, vertically-stacked W-LEDs studied through both experiment and simulation. 

Figure 10a,b show the dependence of the luminance and the CRI on the film configuration, respectively. The luminance is the highest for the DYR configuration whereas it is the lowest for the mixD configuration. In the case of the DYR configuration, the blue light, homogeneously mixed via the diffuser plate, excites the yellow phosphor and the red QD films sequentially. Thus, the yellow peak, which is the most sensitive to the human eyes, is the strongest and the efficiency becomes the highest. On the other hand, the mixD configuration has the diffuser plate on top of the mixed color conversion film. The efficiency of this configuration is generally low due to the low transmission of the diffuser plate. Figure 10b clearly shows that the color-rendering property is closely related to the film configurations. Especially, the individual CRI ‘R9’ (deep red) becomes very low when the red QD film is on top of the yellow phosphor film. The red peak from the QD film in this configuration is generally very low due to insufficient excitation as Figure 9 shows, resulting in low R9 and, thus, Ra and Re. It should be noted from Figure 10b that the highest Ra is still slightly lower than 90 in all cases, which is due to the sharp dip between the blue peak and the yellow peak. This can be improved by adjusting the emitting spectrum of the yellow phosphor as has been experimentally demonstrated and shown in Figure 4. In this case, the CRI value of this high-power LED device is expected to be higher than 90, similar to the experimental results. 

Figure 11a,b show the angular distribution of CCT along the vertical and horizontal directions, respectively. The viewing angle is defined by the angle between the normal direction and the viewing direction, which was changed along the horizontal and vertical directions at a regular interval. It indicates that, in the spherical coordinates, the polar angle is defined as the viewing angle at two fixed azimuthal angle, i.e., 0 and 90° on the emitting surface. Two groups can be recognized from the data sets, one with negligible angular dispersion (YRD, YDR, RYD, and mixD; these configurations will be denoted as Group-I) and the other with substantial angular dispersion (DYR, DRY, Dmix, and RDY; these configurations will be denoted as Group-II). The configurations in Group-I have a common characteristic that the diffuser plate is on top of other color-conversion films except for the YDR configuration. The diffuser plate scatters and mixes the light strongly making the distribution on it Lambertian. Due to the strong diffusing function, color properties look the same irrespective of the viewing angle. In the case of the YDR configuration, the light passing through the top red QD film experiences different optical path lengths causing higher color-conversion probabilities along the higher viewing angles. However, the QD film is very thin, thus the color dispersion caused by the path length difference may be rather small compared to the thick phosphor plate. This is indeed observed from the data shown in Figure 11 where the CCT of the YDR configuration slightly decreases as the viewing angle increases due to the enhanced red component via longer path length in the QD film. On the other hand, all configurations in Group-II show significant color dispersion, i.e., substantially lower CCT values at higher viewing angles. This is mainly due to the large difference in the optical path length in the color-conversion films for the excitation light. As the two films are put over the diffuser plate except for the RDY case, the effect of the path length difference is expected to be pronounced for the light escaping the lighting fixture. In the case of RDY configuration, the color dispersion occurs mainly by the yellow phosphor film, which is thick enough to have a more substantial effect on the angular CCT dispersion compared to the YDR configuration. Finally, we need to remark on some asymmetric behaviors shown in Figure 11. The origin of the slight asymmetry may be most probably due to the insufficient number of rays used in the simulation. Otherwise, two sets of data along both horizontal and vertical directions should have been nearly the same. However, important conclusions derived from the simulation and the resulting design rules described below are not affected by the slight asymmetric behaviors.

The results presented in this study provide several design rules for realizing high-power white LED lighting devices of high color-rendering performances. First, using the red QD film is a very effective way to complement the long-wavelength spectral region and, thus, secure high CRI values. The combination of the yellow phosphor plate and the red QD film confirmed high CRI values of more than 90 in the optimized configuration. The remote design may also assure long-term stability. Second, the color properties, such as color coordinates and CCT, can be tuned in a wide range by changing the order of the two color-conversion components and their thicknesses. These changes have substantial effects on the CCT, CRI, and other color characteristics in addition to efficiency. It gives us the flexibility of tuning CCT values in a wide range by simply changing the order of the components or replacing the component with others having different thicknesses. It indicates that the same lighting fixture can provide quite different color shades in the same space for different lighting purposes; for example, low CCT for bedroom and high CCT for studying room, etc. Finally, the order of the color-conversion components and the diffuser plate is critical in removing the color dispersion of high-power LED lighting devices. Especially, the color-conversion films should be included below the diffuser plate to avoid substantial color dispersion caused by the difference in the optical path length in color-conversion components. 

## 4. Conclusions

A vertically-stacked white LED structure consisting of a yellow phosphor plate and a red QD film was studied through both experiment and optical simulation. The emitting spectra, the color-rendering performance, and the color coordinates were affected significantly by the sequence of the two color-conversion components stacked vertically, as well as the phosphor thickness. It was important to locate the red QD film closer to the blue LED chips to enhance the deep red component and, thus, secure high CRI values of more than 90. The thickness of the phosphor plate had to be optimized between enough color conversion process and excess multiple scattering of light. Based on the consistent results between the experiment and simulation for the small-size, vertically-stacked W-LED, the optimized configuration of high-power, remote-type downlighting white LED devices was studied by simulation. It was shown that the sequence of the two color-conversion components and the diffuser plate was important in removing the color dispersion and that the color-conversion films should be located below the diffuser plate to avoid substantial color dispersion caused by the difference in the optical path length in color-conversion components. The present study shows the design rules of realizing high power, high color-rendering, and remote-type white LED devices that potentially have high long-term stability. 

## Figures and Tables

**Figure 1 nanomaterials-12-02846-f001:**
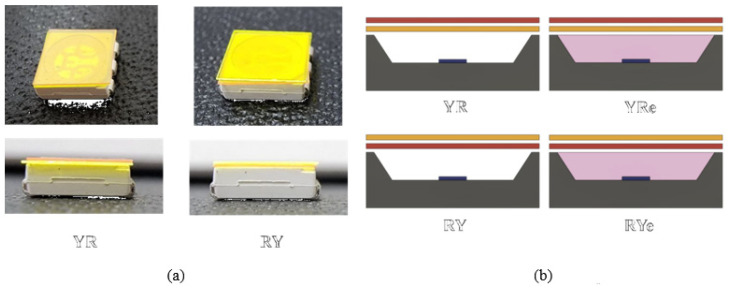
The four vertically-stacked W-LED structures: (**a**) Experimental W-LEDs of YR and RY structures and (**b**) their schematic cross-sectional drawings, including the YRe and RYe structures.

**Figure 2 nanomaterials-12-02846-f002:**
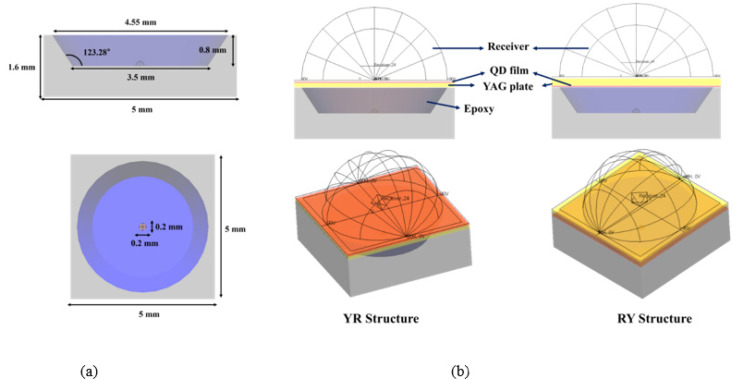
Simulation models for the vertically-stacked W-LED structures: (**a**) the simulation model of the blue LED package and (**b**) the simulation models of the vertically-stacked W-LEDs.

**Figure 3 nanomaterials-12-02846-f003:**
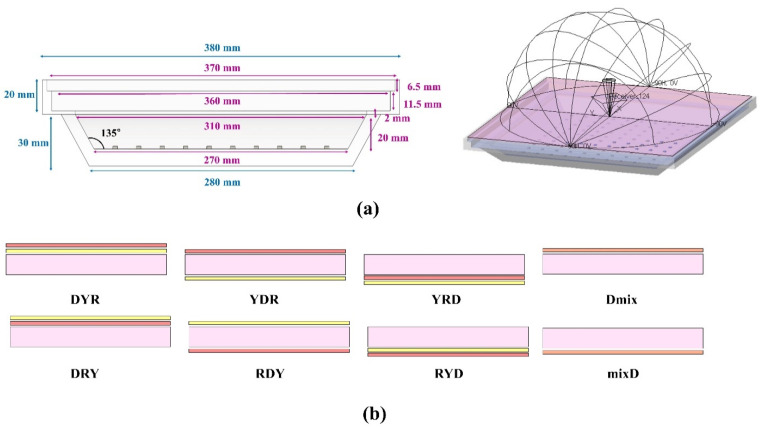
Simulation models of the high-power lighting fixture: (**a**) simulation model of the lighting fixture and (**b**) eight configurations from different stacking of the yellow phosphor film (Y), the red QD film (R), and the diffuser plate (D). “Mix” indicates the film in which both phosphor and QD particles are incorporated together.

**Figure 4 nanomaterials-12-02846-f004:**
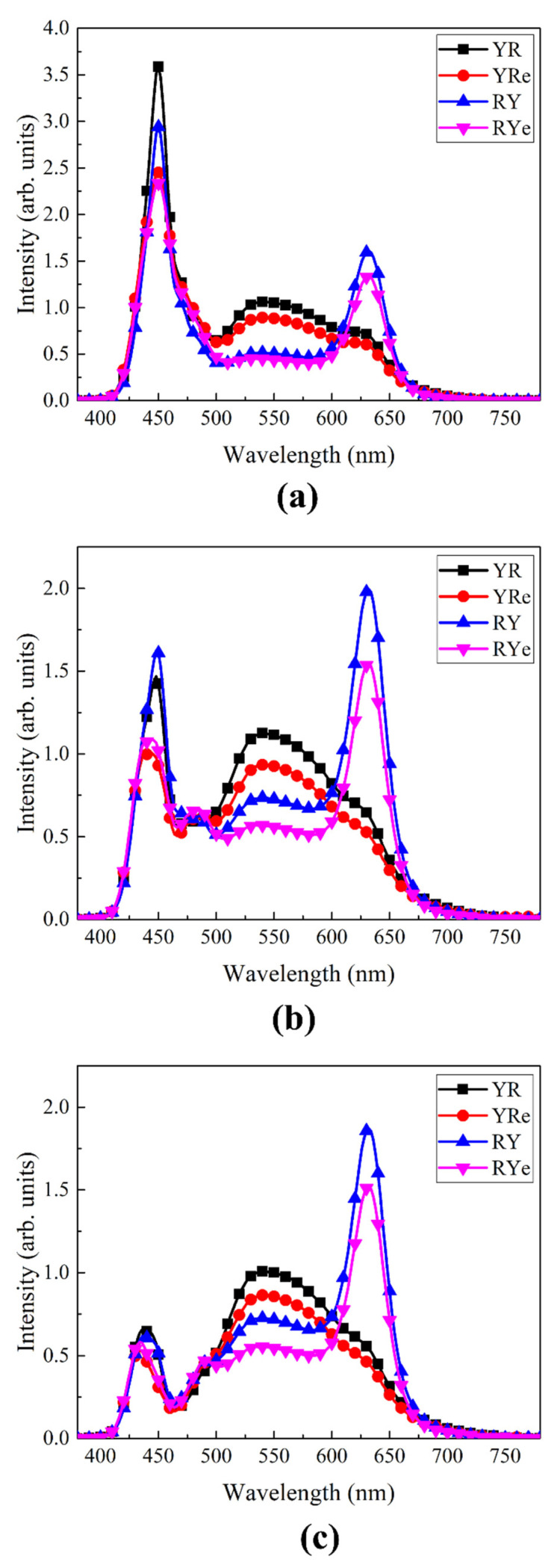
The emitting spectra of W-LEDs for the four configurations at the phosphor thickness of (**a**) 0.1 mm, (**b**) 0.15 mm, and (**c**) 0.2 mm.

**Figure 5 nanomaterials-12-02846-f005:**
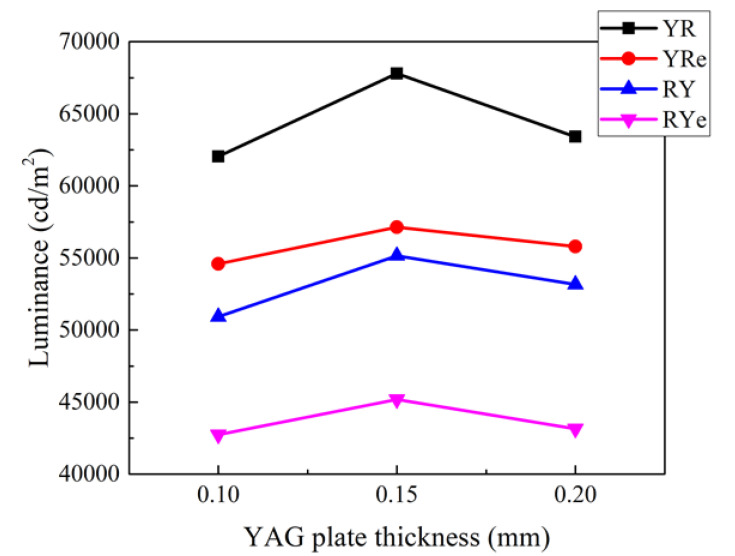
The dependence of the luminance on the thickness of the phosphor plate for the four configurations of W-LEDs.

**Figure 6 nanomaterials-12-02846-f006:**
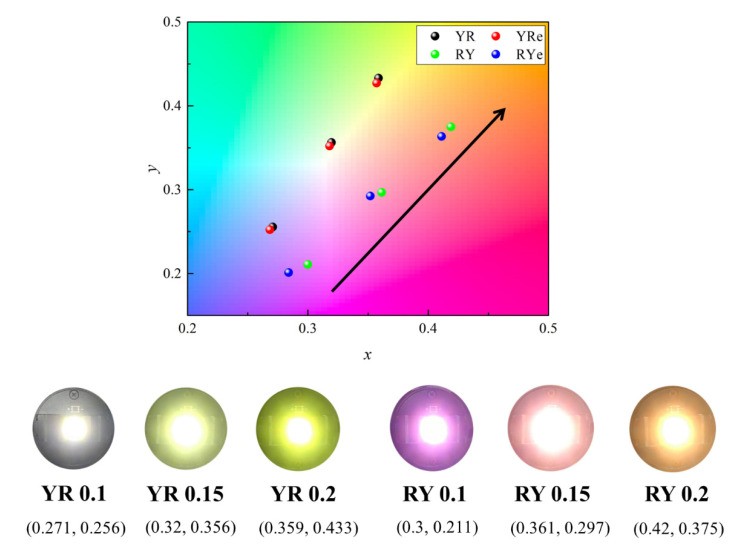
The change in the color coordinates of four configurations on the chromaticity diagram caused by the change in the phosphor thickness. The arrow indicates the direction of increasing phosphor thickness.

**Figure 7 nanomaterials-12-02846-f007:**
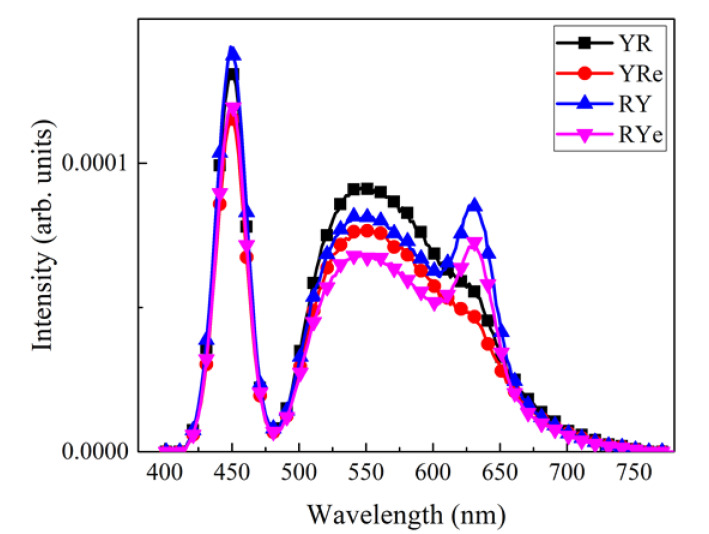
The emitting spectra of four configurations at the fixed phosphor thickness of 0.14 mm in the simulation model.

**Figure 8 nanomaterials-12-02846-f008:**
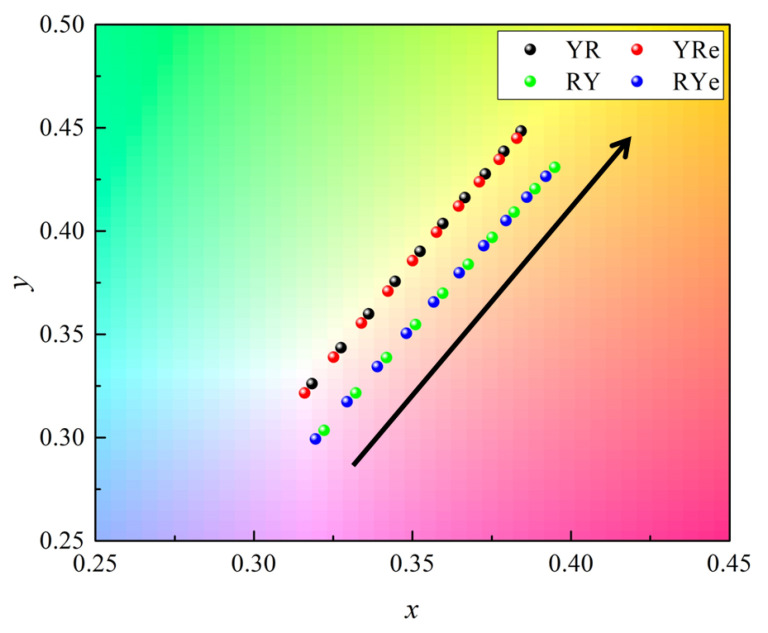
The change in the color coordinates of four configurations on the chromaticity diagram caused by the change in the phosphor thickness in the simulation model. The arrow indicates the direction of increasing phosphor thickness.

**Figure 9 nanomaterials-12-02846-f009:**
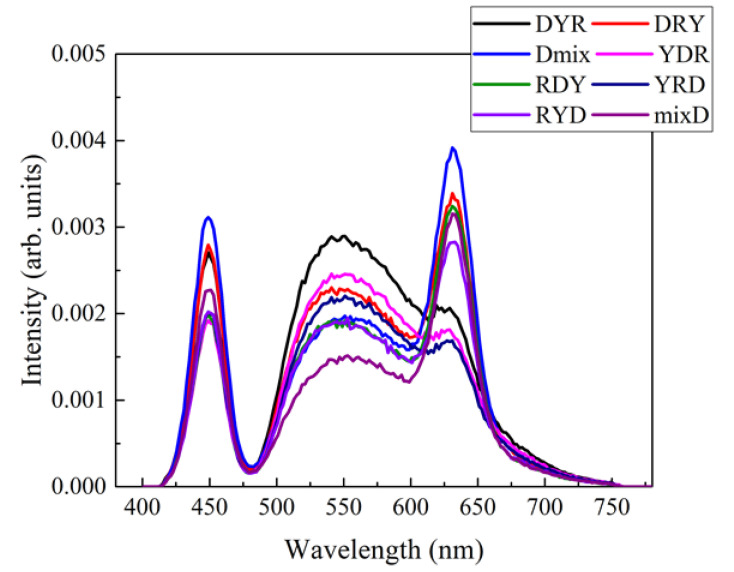
The emitting spectra of eight configurations of the high-power direct-lit W-LED lighting studied by simulation.

**Figure 10 nanomaterials-12-02846-f010:**
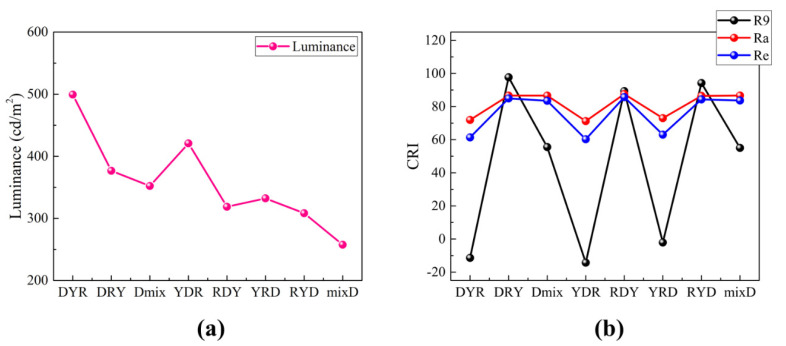
The dependence of (**a**) the luminance and (**b**) the CRI on the film configuration of the high-power direct-lit W-LED lighting studied by simulation.

**Figure 11 nanomaterials-12-02846-f011:**
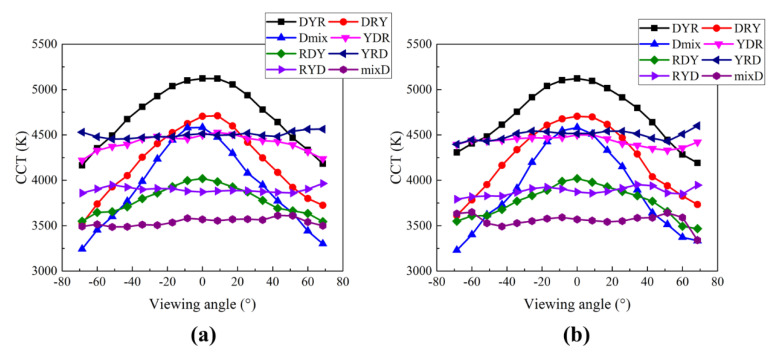
Angular distribution of CCT along (**a**) the vertical and (**b**) the horizontal direction on the emitting surface of the high-power direct-lit W-LED lighting studied by simulation.

**Table 1 nanomaterials-12-02846-t001:** Dimensions and material/optical properties of the W-LEDs.

	Dimension (mm^3^) (Length, Width, Height)	Material	Optical Property
Frame	5.0 ×5.0 × 1.6	PMMA (*n* = 1.4936)	Inside: Gaussian scattering (Sigma 15°, Reflection 85%, Absorption 15%) Outside: Optical absorber
Chip	0.2 ×0.2 × 0.01	GaN (*n* = 2.4)	Fresnel loss
Epoxy	3.5 ×4.55 × 0.8 (Base, Top, Height)	NOA63 (*n* = 1.5637)	
YAG plate	5 ×5 × 0.12~0.21	MgAl_2_O_4_ (*n* = 1.7188)	
QD film	5 ×5 × 0.03	PET (*n* = 1.5733)	

**Table 2 nanomaterials-12-02846-t002:** Dimensions and material/optical properties of the high-power lighting fixture.

	Dimension (mm^3^) (Length, Width, Height)	Material	Optical Property
Frame	380 ×380 × 20 (Top of the Lighting)	Aluminum (*n* = 1.0122)	Inside: Gaussian scattering (Sigma 15°, Reflection 85%, Absorption 15%) Outside: Optical absorber
Diffuser plate	368 ×368 × 2	PolycarbonateLED2045 (*n* = 1.5896), TiO_2_ (*n* = 2.4358)	Fresnel loss
YAG, QD film	368 ×368 × 0.1	PET (*n* = 1.5733)	

**Table 3 nanomaterials-12-02846-t003:** Simulation conditions of the emitting properties of blue LEDs, YAG, and QD particles.

	Emission Peak (nm)	FWHM (nm)	Weight Percent (wt%)	Mean Free Path (mm)
LED	450	25	-	-
YAG	540	100 (below 540 nm) 170 (above 540 nm)	16 (W-LED) 50 (Lighting)	-
QD	632	30	-	0.01 (W-LED) 0.12 (Lighting)

## Data Availability

Data presented in this article is available on request from the corresponding author.

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
