# Peer review of "Structural Optimization of Vertically-Stacked White LEDs with a Yellow Phosphor Plate and a Red Quantum-Dot Film"

_nanomaterials, 2022, doi:10.3390/nano12162846_

Round 1
Reviewer 1 Report
In this manuscript, the authors have studied a remote-type white LED with red QD film and yellow phosphor plate from both experiment and simulation. By changing the stacking sequence and structure of the QD and phosphor plate, they have optimized the emission characteristic of the LED (e.g., color rendering index). Also, the experimental results have been found to be consistent with the simulated data. In general, this is an interesting work. Here the reviewer only has a few technical questions.
1. It is great to use the simulation results as a comparison with the measured data. But in Fig.9-11, it seems that they are all from simulation. If experimental data can also be measured and compared with them, it could further enhance the clarity and influence of the paper.
2. In the introduction part, the authors have emphasized that one of the major issues of the W-LED is the long-term stability due to the material degradation and the remote design is useful in order to mitigate this problem. Therefore, the LED in this manuscript also adopts this remote-type structure. Could the authors experimental test the long-term stability of these LEDs, including YR, RY, YRe, and RYe?
Author Response
We highly appreciate the reviewer’s favorable comments.
We revised our manuscript according to the comments as follows. The changes were colored in red in the revised version.
<Reviewer 1>
Comment(1): It is great to use the simulation results as a comparison with the measured data. But in Fig.9-11, it seems that they are all from simulation. If experimental data can also be measured and compared with them, it could further enhance the clarity and influence of the paper.
- Reply: That’s right. The data shown in Fig. 9-11 are about optical properties of high-power downlighting LED devices, which we studied by only simulation. The purpose of this last study was to utilize the results obtained from the small-size, vertically-stacked LEDs to optimize large-size downlighting devices. We agree with the reviewer in that experimental tests are necessary to confirm the simulation result, which is beyond the scope of the present study and will be investigated in near future to be reported in a separate study.
Comment(2): In the introduction part, the authors have emphasized that one of the major issues of the W-LED is the long-term stability due to the material degradation and the remote design is useful in order to mitigate this problem. Therefore, the LED in this manuscript also adopts this remote-type structure. Could the authors experimental test the long-term stability of these LEDs, including YR, RY, YRe, and RYe?
- Reply: In principle, the long-term stability test is very important and possible by turning on the LED and monitoring the optical properties for a long time. However, it usually takes more than 1000 hours even we try to estimate the lifetime by extrapolation based on theoretical models predicting the lifetime of lamps. Thus, it would not be plausible to carry out this important test within the deadline of revision. It should be done separately, which we hope to report in future.
Reviewer 2 Report
This is an interesting study of the design of white LED based on a blue GaN LED. While the underlying physics is quite obvious, the merit of the paper is to provide quantitative analyses of the efficiency and color parameters of the WLED. Hence this paper will be useful and derserve publication. I have the following comments which could be addressed: 1-Page 11 : thicker phosphor thickness is not correct. larger thickness is preferred 2-Simulations and experimental results overall agree. However, there are some discrepancies which deserve comments: 2a-the red emission is lower in the simulation of RY and Rye, than in experiment 2b-optimum of phosphor is found to be 0.15 mm while simulation predicts below 0.12mm. 3-What is the polarization of the blue light ? it is supposed to be mainly TE polarized. Was it taken into account ? 4-Figure 11: viewing angle along the vertical direction: do you mean the theta angle, where theta is the angle between the normal to the surface and the viewing angle ? Horizontal direction ? do you mean theta close to 90° and you calculate as a function of the Phi angle ? Why is fig11a not symmetrical for +/- theta for all configurations ? RYD for instance is not symmetrical
Author Response
We highly appreciate the reviewer’s favorable comments.
We revised our manuscript according to the comments as follows. The changes were colored in red in the revised version.
<Reviewer 2>
Comment(1): Page 11 : thicker phosphor thickness is not correct. larger thickness is preferred.
- Reply: We corrected this mistake on the same page.
Comment(2): Simulations and experimental results overall agree. However, there are some discrepancies which deserve comments: 2a-the red emission is lower in the simulation of RY and Rye, than in experiment 2b-optimum of phosphor is found to be 0.15 mm while simulation predicts below 0.12mm
- As the reviewer correctly indicated, the experimental and simulation results show some discrepancies. This is because the simulation conditions are not exactly the same with the experimental conditions. We thus included further explanation on this point as follows.
(page 7, second paragraph) “One remark on the spectrum is that the red peaks in the emitting spectra of RY and RYe configurations are lower than those of the corresponding experimental configurations. The main origin of this difference is directly related to the MFP of red QDs in the film. If the MFP is finely adjusted, the emitting spectrum of the simulation would be more similar to that of the experiment.”
(page 8, first paragraph) “The difference in the optimal thickness between the experiment and simulation is due to the fact that the simulation conditions are not exactly the same with the experimental conditions. The phosphor plate used in the experiment is a ceramic-type where grain-boundary effect is also included, which is difficult to include in the simulation.”
Comment(3): What is the polarization of the blue light ? it is supposed to be mainly TE polarized. Was it taken into account ?
- The emitted light of blue LED is in general unpolarized because it is not a coherent source. Therefore, both the blue light from LED chips and the white light from the LED package are unpolarized.
Comment(4): Figure 11: viewing angle along the vertical direction: do you mean the theta angle, where theta is the angle between the normal to the surface and the viewing angle ? Horizontal direction ? do you mean theta close to 90° and you calculate as a function of the Phi angle ? Why is fig11a not symmetrical for +/- theta for all configurations ? RYD for instance is not symmetrical
- The definition of the viewing angle was added in the second paragraph of page 9 as “The viewing angle is defined by the angle between the normal direction and the viewing direction, which was changed along the horizontal and vertical directions. It indicates that, in the spherical coordinates, the polar angle is defined as the viewing angle at two fixed azimuthal angle, i.e., 0 and 90o on the emitting surface.”. Regarding the slight unsymmetrical shape, we included the following sentences on page 9 (second paragraph). “Finally, we need to remark on some asymmetric behaviors shown in Fig. 11. The origin if the slight asymmetry may most probably be due to the insufficient number of rays used in the simulation. Otherwise, two sets of data along both horizontal and vertical directions should be nearly the same. However important conclusions derived from the simulation and the resulting design rules described below are not affected by the slight asymmetric behaviors.”
We hope the present version be enough for our manuscript to be accepted for publication in this journal.